# Antioxidative and Antimicrobial Evaluation of Bark Extracts from Common European Trees in Light of Dermal Applications

**DOI:** 10.3390/antibiotics12010130

**Published:** 2023-01-09

**Authors:** Sissy Häsler Gunnarsdottir, Lukas Sommerauer, Thomas Schnabel, Gertie Janneke Oostingh, Anja Schuster

**Affiliations:** 1Biomedical Sciences, Salzburg University of Applied Sciences, Urstein Sued 1, 5412 Puch, Austria; 2Department of Forest Products Technology & Timber Constructions, Salzburg University of Applied Sciences, Markt 136a, 5431 Kuchl, Austria; 3Salzburg Center for Smart Materials, c/o Department of Chemistry and Physics of Materials, Paris Lodron University of Salzburg, Jakob-Haringer-Straße 2a, 5020 Salzburg, Austria; 4Department of Material Sciences and Process Engineering, Institute of Physics and Materials Science, University of Natural Resources and Life Sciences, Peter-Jordan-Straße 82, 1190 Vienna, Austria; 5Faculty of Furniture Design and Wood Engineering, Transilvania University of Brasov, B-dul. Eroilor nr. 29, 500036 Brasov, Romania

**Keywords:** European wood bark extracts, extraction technologies, antioxidative potential, antimicrobial potential, dermatology

## Abstract

Plant species have developed effective defense strategies for colonizing diverse habitats and protecting themselves from numerous attacks from a wide range of organisms, including insects, vertebrates, fungi, and bacteria. The bark of trees in particular constitutes a number of components that protect against unwanted intruders. This review focuses on the antioxidative, dermal immunomodulatory, and antimicrobial properties of bark extracts from European common temperate trees in light of various skin pathogens, wound healing, and the maintenance of skin health. The sustainability aspect, achieved by utilizing the bark, which is considered a byproduct in the forest industry, is addressed, as are various extraction methods applied to retrieve extracts from bark.

## 1. Introduction

The bark of a tree is a functional material for the protection of living parts against biotic and abiotic influences from the environment. After the harvesting of trees the bark is the packaging of the wood and guards the materials from transport damage. Before the wood is processed the trunk will be debarked, and after that two material fractions (wood and bark) are separately manipulated for various uses [1].

Wood is used by humans for multiple purposes. These applications include the use of wood for heat generation [2], as a building material [3], as a basis for the paper industry [4], and to build furniture [5]. Nevertheless, bark is regarded as an interesting material that can be used for a range of different purposes [6]. The bark of some trees was found to contain compounds that can be extracted and used in several medical and cosmetic applications [6]. This use is mainly applied in the field of alternative medicine and is more common in non-European continents due to a longer history and greater acceptance of alternative medicine (for example, Chinese alternative medicine) [7]. In order to make it more attractive to use different parts of European trees for medical and cosmetic applications, knowledge of the constituents and biological effects of these parts of trees need to be defined in greater detail. This review aims at summarizing the available knowledge on the potential of European wood-species-derived bark extracts for their uses in biomedicine.

Specific components of trees have the potential for medical and cosmetic applications. The use of tree-derived extracts or infusions has been applied for centuries in the field of alternative medicine. The Mayan culture boiled the wood of *Guaiacum officinale* and *Guaiacum sanctum* for the treatment of the venereal disease syphilis [8]. In more recent years, the effects of resins extracted from *Guaiacum* have been studied for anti-inflammatory, antitumor, and antirheumatic effects [9,10]. Another species of wood that has been known and used in pharmacy for a long time is oak (*Quercus*) [11]. All parts of the plant contain tannins, whereas the bark is the main focus of pharmaceutical and folk medicine [12]. In the form of resins, poultices, or gargle solutions, it is used for its antiseptic effect in the treatment of inflammatory diseases of the mucosa in the genital, anal, and oral areas [11]. Furthermore, the food supplement Robuvit^®^ (Geneva, Switzerland) finds its application in fatigue symptoms, muscle and joint pain, and sleep disorders. Robuvit^®^ is a capsule that contains the extract of French oak wood, *Quercus robur*. Various studies have shown that taking Robuvit^®^ significantly reduces heart rate, improves alertness and sleep patterns, and significantly reduces oxidative stress compared to the control group [13,14,15].

In addition, tree extracts are also used in classical medicine. This can be illustrated by the original extraction of salicin from the bark of the white willow (*Salix alba*), which was the basis of today’s production of acetylsalicylic acid [16]. Although the use of salicin is nowadays reduced due to the high number of side effects in the gastrointestinal area, the effect of salicis cortex extract on platelet aggregation, osteoarthritis, and rheumatoid arthritis is still being investigated [17,18,19]. Another example is the use of birch bark extracts in the field of wound healing [20,21,22]. The resulting product was named Episalvan^®^ (Niefern-Öschelbronn, Germany) and was approved by the European Medicines Agency (EMA) in 2016, however is no longer licensed; Fisuvez^®^ (Niefern-Öschelbronn, Germany), a dry extract from birch bark using n-heptane for the treatment of epidermolysis bullosa, was approved in 2022 [23].

This leads to the assumption that there will be a number of other European tree species containing compounds that can be of use for medicine and cosmetic applications. Tree-derived extracts do thereby always consist of a variety of different compounds, and it is well-known from the field of mixture toxicity [24] that the biological response against a mixture of chemicals is not always just the sum of the responses against single compounds. Additive, inhibitory, and synergistic effects play major roles in biological responses and should thus be taken into consideration [25]. In addition, from other biological fields it is known that replacing a biologically derived agent (for example, serum in cell culture experiments or blood for transfusions) with a chemically produced product (serum replacements or synthetic blood) does not lead to the same biological response [26]. Therefore, in this review we focus on studies that investigated the extracts of tree bark as a whole, where a characterization of a single compound plays an important role, in current and future studies, in order to make these extracts acceptable for use in biomedicine. Currently, the exact contents of such extracts are not fully defined. Next to the partly unknown constituents of extracts obtained from trees or parts of trees, the levels of these compounds will also differ between different trees of a specific species. These differences are caused by the speed of growth and the environment in which the trees grow (height, pollution, soil, etc.) [27,28]. Knowledge of these differences will be of significant importance to enabling a standardization process or at least to predict the potential effects of single wood extracts based on different growth-related circumstances.

Tree-derived extracts can be used in a number of applications, as was illustrated above. Currently, several studies have focused on dermal applications [20,29,30,31,32]. The reason for this is likely to be manifold. In a number of skin-related disorders infection with microorganisms, oxidative stress, and inflammation play a dominant role, often leading to poor healing processes or chronic wounds [33]. Various plant-derived extracts are known to possess strong antioxidative potentials in addition to antimicrobial and immune-regulating activity that might be beneficial for a number of skin-related disorders [34]. There are a number of skin-related diseases that are, in mild to moderate cases, not life-threatening to an individual who is affected; however, the long-term effects can be major, since infections and major scarring can occur, and, due to the latter, the mental harm can be significant as well [35]. Acne vulgaris is an example of such an infectious skin disease. Normally, good skincare can prevent or improve the symptoms of this disease. However, if normal skincare is not sufficient, the use of antibiotics and/or hormonal preparations is often applied for treatment. Since it is predominantly young people that are affected, the use of hormones should on all accounts be avoided in this patient population, who are still in a developmental life-stage. In addition, due to the general rise in the use of antibiotics and the problems with the generation of antibiotic-resistant bacterial strains, the prescription of this medication should be generally reduced. Therefore, alternative treatment methods, for example by using tree-derived extracts that are antibacterial, anti-inflammatory, and antioxidative, as well as those that stimulate wound healing, might be a solution for solving some of these problems.

Within this review, the latest literature available on extracts from bark of common European trees is discussed in the light of available extraction technologies, their antioxidative potential as well as antibacterial and skin-related wound healing properties. Although close care was taken to perform an inclusive literature review, it can, based on the available sources, not be completely ruled out, that single manuscripts are not included. As part of this review, the potential use of bark-extracts for dermatological applications will be discussed.

There is an ongoing debate as to whether crude plant extracts might have greater biological activity than isolated constituents at an equivalent dose. The aim of this article is to review the knowledge on the crude extract of tree bark with the notion that the variety of substances present in a crude extract work synergistically to counter microbial growth, mediators of oxidative stress, and imbalanced immune responses in the light of dermatological disorders.

## 2. Occurrence of Common Temperate Trees and Bark Utilization in the Circular Economy

Compared to other continents, Europe possesses the largest area of forests, with 1.005 million hectares according to the Earth Policy Institute [36]. European forests consist mainly of Scots pine (*Pinus sylvestris*)—31%, Norway spruce (*Picea abies*)—21%, and beech (*Fagus sylvatica*)—7.1%, followed by species such as birch (*Betula pubescens*), oak (*Quercus robur*), and other pine species (Pinus)—< 5% [37]. Fir (*Abies alba*) accounts for only about 1.5% [37]. Table 1 summarizes the plant species discussed in this review by listing the botanical classification and most common species of each genus within Europe for which sufficient literature could be found. In the following text the common name of the species will be stated to facilitate ease of reading.

Approximately 115 tree species are identified in Europe, with varying levels of importance in terms of their usage as building materials, furniture, flooring, etc. In particular, the species spruce, fir, and pine play important roles due to their use as building materials [38]. The amount of wood utilization from these species is large and leads to a high amount of residual materials, such as tree bark. Depending on the growing area, species, and other environmental factors, bark tissue accounts for about 10–20% of wood biomass. For example, the amount of bark available annually in 2018 was estimated to be 23,000,000 m^3^ based on a survey of agroforest residue potential in the European Union. Thus, tree bark, as a residual material, especially from the production of pulp or wood-based materials, is available in large quantities and is mainly used for energy production [39]. New fields of added-value applications have already been explored for bark biomass, such as their use in board materials, as an insulating material, in the form of filler material in biogenic plastics, or as building blocks for further processing [1,6,40]. These approaches particularly address the aspect of the circular economy in order to use the residual material as a new product. Nevertheless, when considering the composition of wood and bark biomass, it can be seen that bark biomass, in addition to having a higher lignin content, also has a higher content of extractives, which makes this material particularly interesting for the extraction of valuable compounds [41,42]. Thus, bark biomass contains a wealth of components that can be used for high-value products [6,43].

## 3. Extraction Technologies in Relation to and Relevant for Biomedical Applications

To retrieve the components from wood bark, several extraction technologies were developed. Maceration, infusion, steam distillation, hydrodistillation, and Soxhlet extraction are considered as conventional extraction techniques [44]. More advanced approaches include ultrasound- and microwave-assisted extraction, pressurized and supercritical fluid extraction, ionic liquids and deep eutectic liquids, and enzyme-assisted extractions [12,43,45]. In the case of maceration, the extraction material and a solvent are dispersed several times, often while agitated. Afterwards, the solid residues are filtered out to receive the dispersed substances [46]. This process is gentle and simple, but has limited extraction effectiveness. In the case of infusion, heated solvents below boiling point are used to extract small amounts of material. Thermolabile compounds can be degraded due to the higher solvent temperature [47]. For the steam distillation and hydrodistillation techniques, heat causes a breakdown of the extraction material, thus releasing the desired substances. For steam distillation, boiling water passes directly or indirectly through the extraction material. In hydrodistillation, the plant material is placed in the same extraction vessel as the water. Both distillation methods use steam to carry the desired substances to a condenser to obtain bioactive compounds. If essential oils or volatile compounds are desired, steam distillation is the most suitable method [48]. Soxhlet extraction makes use of hot solvents, usually with a low boiling point, that are continually heated and refluxed [44]. Disadvantages of this technique are thermal degradation due to relatively high heating temperatures and high solvent consumption [49]. In ultrasound-assisted extractions, ultrasonic waves cause continuous expansion and compression inside the extraction material [50]. The solvents can penetrate deeply into the extraction material and the higher contact surface leads to higher extraction effectiveness. As the name already references, in microwave-assisted extraction electromagnetic waves with high frequencies are used to infuse changing dipole orientations, which leads to vibration and the heating of the material [51].

Pressurized fluid extraction makes use of high pressure, which is applied to keep the solvent in a liquid state under elevated temperatures [52]. This process leads to shorter extraction times and requires lower solvent amounts compared to conventional systems. However, the high temperatures applied can lead to the thermal degradation of the extracted substance. In supercritical fluid extraction, the pressure and temperature are even higher; the solvent reaches its supercritical state, where it behaves like a liquid and gas simultaneously [53]. In most cases, CO_2_ is used due to its low critical point at moderate temperatures and pressure conditions [54]. Polar compounds cannot be extracted efficiently with CO_2_ [48].

Since environmental aspects are of increasing importance, ionic and deep eutectic liquids are seen as promising agents to replace conventional organic solvents. In ionic liquids, mixtures of salts are in a liquid state at room temperature due to fitting anion–cation pairs [55]. The hydrogen bonds of constituted biopolymers, such as cellulose, lignin, and chitin, are disintegrated, and this leads to the liberation of cell constituents [56]. Mixtures of two or more solid components can form a deep eutectic solvent (DES). The resulting mixture with a melting point below the melting points of the individual components can be obtained [57,58]. This approach is environmentally friendly due to its biodegradability, sustainability, low toxicity, recyclability, and low volatility. Further advantages are simplicity, low costs, and high extraction efficiency [59]. In enzyme-assisted extraction, enzymes such as celluloses, hemicelluloses, and pectinase are used to break down the structural integrity of plant cell walls to make the cell itself accessible for solvent penetration and the liberation of constituents [60]. This is a promising approach for removing organic solvents, although costs are still a limiting factor [61,62].

In the light of biomedical applications, rather gentle extraction technologies or extraction parameters are favorable due to the better preservation of the biological active substances. In addition, biological or biocompatible solvents, preferably water, facilitate their use for cosmetic and medical applications. In all of the extraction methods described above, direct comparison remains difficult due to different extraction parameters. The sample matrix, particle size, temperature, pressure, time, solid-to-solvent ratio, and solvent system have a strong influence on extraction efficiency and effectiveness [63]. Higher yields can be achieved if the particle size of the bark is reduced, without influencing the types of compounds extracted. This can be attributed to the increased active surface area and enhanced contact of the extraction material with the solvent [64]. Furthermore, finer grinding leads to a higher degree of destroyed cells, which also increases the extraction yield. Increasing the solid-to-solvent ratio ensures the prevention of saturation in the solvent and therefore increases the possible extraction yield [65]. The use of microwave-assisted extraction enhances the extraction yield of maritime pine bark, without changing the structure, and the reactivity of the obtained extractives in comparison with hot water extraction [64]. Microwave-assisted extraction was shown to be more efficient in comparison to conventional extraction techniques in terms of antioxidant activity in ferric reducing antioxidant power (FRAP), oxygen radical absorbance capacity (ORAC), and total phenolic content (TPC) [66]. With regard to the yield and total phenol content of bark of Monterey pine (*Pinus radiata*), Soxhlet extraction had the highest performance, followed by microwave-assisted extraction, ultrasound-assisted extraction, and conventional maceration. Furthermore, an increased extraction time led to thermal degradation for Soxhlet extraction [67]. A detailed description of the different extraction methodologies, including various variable parameters in relation to total yield, total phenolic content (TPC), and radical scavenging ability (DPPH), for an excerpt of tree species discussed in this review is given in Table 2. An extensive list, including 11 tree species, can be found in Appendix A.

In summary, it is well-known that the extraction method used plays a fundamental role in terms of the yield of the various compounds present in wood bark extracts. This further influences the biological activity of these extracts. Therefore, a thorough characterization of the compounds present in the extracts and a comparison of various extraction methods are fundamental for understanding the biological activity and identification of the most suitable extract relating to the desired application.

## 4. Antioxidative Effects of Bark Extracts

Reactive oxygen species (ROS) are continuously generated in cells during metabolism and play a crucial role in normal cellular function, as well as in immune responses [68] and the biosynthesis of molecules [69]. Their quantity is regulated by protective cellular defense mechanisms, namely by detoxifying these agents. However, if ROS generation exceeds the protective intracellular defenses, a condition termed oxidative stress occurs. Exaggerated reactive species can cause cellular damage due to DNA lesions, lipid peroxidation in the membranes, and the oxidative modification of proteins, in addition to disrupting vital cellular processes and increasing mutations [70]. This state can contribute to the pathophysiology of a number of chronic and degenerative diseases. Pathological conditions or immune reactions are endogenous causes of ROS generation, whereas sunlight (UV radiation), pollution, and lifestyle risk factors, such as an unhealthy diet [71] and cigarette smoking [72], are external causes of exaggerated ROS accumulation.

An antioxidant is any compound that inhibits oxidation either by removing potentially damaging oxidizing agents or delaying/preventing oxidative damage [73], therefore playing an important role in the protection against the consequences of oxidative stress. Antioxidant defenses can be exogenous or endogenous, the latter being categorized into enzymatic or non-enzymatic antioxidants [74]. Cells harbor a number of antioxidant enzymes, including superoxide dismutase (SOD), glutathione peroxidase (GPx), and catalase (CAT) [74], as well as non-enzymatic antioxidants such as metal-binding proteins (including albumin, ferritin, and myoglobin), glutathione, and coenzyme Q10 [75]. Plants are a rich source of biologically active substances that can have a significant effect on oxidative-stress-related damage. There are a variety of natural antioxidants that can be provided through foods and/or dietary supplements or as a part of medication, such as vitamin C, vitamin E, carotenoids, or phenolic compounds [76]. Polyphenols are secondary metabolites present in all parts of plants, with flavonoids, phenolic acid, and tannic acid (especially condensed tannins—proanthocyanidins) being among the most essential compounds with regard to antioxidant activity [77]. Procyanidins, a group of the proanthocyanidin class of flavonoids, can represent up to 50% of polyphenols in tree barks, being the most abundant polyphenols after lignin, with pine as one of the plants with the highest content of these [78].

**Table 2 antibiotics-12-00130-t002:** Total phenolic content (TPC) and free radical scavenging ability (DPPH*) of tree species extracts in relation to extraction methodologies and parameters.

Species	Extraction Methodology *; Extraction Tissue; Pretreatment; Solvent; Solid:Solvent Ratio; and Extraction Parameters	TPC	DPPH	Ref.
Black alder	ME; whole bark; drying 60 °C 24 h, ground, 0.5 mm; H_2_O; 1:100; and 60 min, 160 °C	29 mg GAE */g	276.6 mg AAE */g	[79]
	UAE; bark; drying 40 °C 76 h, ground, 0.2–0.63 mm; EtOH/H_2_O 80/20; 1:100; and 15 min, 74 °C, 150 W	21.25 mg GAE/g	5.7 μg/mL (IC_50_)	[80]
European white birch	PE; bark; unknown; H_2_O; 1:8; 180 min, 200 °C	440.74 mg GAE/g	81.94 AA%	[81]
	UAE; bark; drying 40 °C 76 h, ground, 0.2–0.63 mm; EtOH/H_2_O 80/20; 1:100; and 15 min, 74 °C, 150 W	29.2 mg GAE/g	7.5 μg/mL (IC_50_)	[80]
Common beech	ME; inner bark; deactivation w. microwave, 2 min, 700 W; ground; EtOH/H_2_O 80/20; 1:100; and 5 h, 25 °C, stirring	48.3 mg GAE/g	13 μg/mL (IC_50_)	[82]
	PE; bark; unknown; H_2_O; 1:8; 180 min, 200 °C	297.86 mg GAE/g	50.68 AA%	[81]
	UAE; inner bark; deactivation w. microwave, 2 min, 700 W; ground; EtOH/H_2_O 80/20; 1:100; and 30 min, 25 °C	44.49 mg GAE/g	8.4 μg/mL (IC_50_)	[82]
	MAE; inner bark; deactivation w. microwave, 2 min, 700 W; ground; EtOH/H_2_O 80/20; 1:100; and 20 min, 120 °C	65.22 mg GAE/g	13 μg/mL (IC_50_)	[82]
European larch	ME; bark; drying, ground, 0.5 mm; EtOH/H_2_O 50/50; 1:8; and 94 min, 58 °C	10.79 mg GAE/g	14.25 mg TE */g	[83]
	PE; bark; unknown; H_2_O; 1:8; 180 min, 200 °C	277.62 mg GAE/g	48.35 AA%	[81]
	UAE; bark; drying, ground, 0.5 mm; EtOH/H_2_O 50/50; 1:8; and 94 min, 65 °C	6.26 mg GAE/g	7.89 mg TE/g	[83]
	MAE; bark; drying, ground, 0.5 mm; EtOH/H_2_O 50/50; 1:8; and 62 min, 100 W	10.7 mg GAE/g	14.59 mg TE/g	[84]
Common oak	UAE/MAE; bark; drying, ground, 0.5 mm; EtOH/H_2_O 50/50; 1:10; and 2 min, 40 rpm, 300 W, 100% amplitude	596 mg GAE/g	838 mg TE/g	[79]
	ME; whole bark; drying 60 °C 24 h, ground, 0.5 mm; H_2_O; 1:100; and 60 min, 160 °C	18.09 mg GAE/g	171.5 mg AAE/g	[80]

* 2,2-Diphenyl-1-picrylhydrazyl (DPPH); maceration (ME); ultrasound-assisted extraction (UAE); microwave-assisted extraction (MAE); pressurized extraction (PE); ascorbic acid equivalents (AAEs); Trolox equivalents (TEs); and gallic acid equivalents (GAEs).

There is proven evidence that there is a correlation between high levels of total phenols and the antioxidant capacities of plant extracts [85,86,87]. Due to their redox properties, polyphenols can contribute in manifold ways towards the defense against free radicals, acting as reductants, radical scavengers, hydrogen donators, and metal chelators [88]. Phenolic compounds also exhibit multiple functions related to the maintenance and reparation of DNA, cell differentiation, the deactivation of pro-carcinogens, and other cellular actions [78].

A variety of techniques are used to study the ROS-mediated damage and antioxidative capacity of different tree barks. A review by Kumar et al. summarizes a number of antioxidant activity assays and methods, including the ones mentioned below [89]. Some of the most commonly used methods for evaluating the antioxidant activities of bark extracts are chemical-based assays, exploiting the scavenging activity, reducing capacity, or inhibition of lipid peroxidation of potential antioxidants. DPPH (2,2-diphenyl-1-picrylhydrazyl) scavenging assays are one of the most widely employed methods. The reaction of DPPH with an antioxidant leads to a color change in the media; the absorbance can be measured and the antioxidant capacity expressed as IC_50_ values, for example, in µg/mL. In addition, ABTS (2,2′-azino-bis(3-ethylbenzothiazoline-6-sulfonic acid) assays are often used. A sample’s ability to scavenge the ABTS radical cation is compared to Trolox, a standard antioxidant and, e.g., expressed in millimoles of Trolox equivalents (TE). FRAP (ferric antioxidant power reduction) assays, CUPRACs (cupric ion reducing power assays), and the Folin–Ciocalteu method (total phenolic content assay) are based on the ability of antioxidants to accept electrons from, for example, transition metals (notably iron and copper) and therefore acting as reducing agents.

Further methods to determine the extent of oxidative damage in cells are the detection of lipid peroxidation, protein carbonylation, or the evaluation of the glutathione ratio. Lipid peroxidation proceeds by three distinct mechanisms, wherein free-radical-mediated oxidation plays a major role. Various assays are available for measuring the level of lipid oxidation products in human cell lines using different lipid peroxidation products as readouts. The thiobarbituric reactive substances (TBARS) method evaluates the reaction of a product of unsaturated lipid degradation (malondialdehyde—MDA) with thiobarbituric acid (TBA). Due to ROS-induced lipid degradation, MDA is formed, a reactive species causing toxic stress in cells and acting as a marker for oxidative stress. Protein carbonylation is a major hallmark of oxidative damage and by far the most commonly used marker of protein oxidation. Various mechanisms can lead to the formation of carbonyl groups on protein side chains by ROS, which are summarized in the review by Dalle-Donne et al. [90]. Most assays are based on the derivatization of the carbonyl group with 2,4-dinitrophenylhydrazine (DNPH), which leads to the formation of a stable DNP hydrazone product [91]. Reduced glutathione (GSH), a ubiquitous tripeptide thiol, is a vital intracellular and extracellular protective antioxidant that plays a number of key roles in the control of signaling processes. Within a resting cell, the ratio of reduced to oxidized glutathione exceeds 100:1; it can decrease to 10:1 or even 1:1 during oxidative stress, and it can therefore be used as a marker for cellular toxicity [92]. Various assays exist that quantitatively measure the amount of GSH or GSSG (glutathione disulfide) based on a fluorometric method. The modulation of enzyme activity, including phospholipase A2 (PLA2), cyclooxygenase (COX 1/2), lipoxygenase (5-LOX), and inducible nitric oxide synthase (iNOS), can be measured by using enzyme activity assay kits based on colorimetric detection. Chemical assays and in vitro cellular models are mostly used to detect cellular antioxidant activity since they are more effective compared to time-consuming, low-throughput, and high-cost in vivo models [93].

The search for natural antioxidants has received a lot of attention lately, and tree bark has been found to be a rich natural resource for active antioxidant compounds. Dudonné et al. compared the total phenolic contents and antioxidative properties of 30 plant extracts from different parts of plants with DPPH, ABTS, ORAC, FRAP, and SOD assays, with oak, pine, and cinnamon bark extracts proving to have the highest antioxidant capacities as well as the highest phenolic contents [86]. A number of studies have shown that tree bark has higher levels of total phenols and presumably antioxidative activities compared to other parts of the same plant [94,95,96,97]. Several techniques have been developed to obtain phenolic compounds from tree bark, differing in their mechanisms and the solvents used. This leads to different results in terms of yield, total phenolic content, and antioxidant activity, as has already been confirmed by several studies. A study used low-frequency ultrasound combined with different solvents to intensify extraction from the whole bark of 10 common wood species from Hungary and determined the total antioxidant capacity of the extracts. Aqueous 80% ethanol was more efficient than aqueous 80% acetone for antioxidant extraction in all of the assays examined, except DPPH. TPC, DPPH, ABTS, and FRAP showed the oak extract to have the highest antioxidant capacity, followed by European ash and black locust [80]. A further study compared the total phenolic and total flavonoid extraction yields of water vs. an ethanol–water mixture (60% ethanol) in three different oak species, showing that the extraction yields were higher for the ethanol–water solution, particularly regarding TFC (total flavonoid content), probably due to a higher solubility of flavonoids in alcohol [98]. García-Pérez et al. investigated crude extracts from different Canadian wood species obtained by maceration or hot water extraction, with yellow birch extract presenting the highest contents of polyphenols obtained by maceration and black spruce extract presenting the highest contents of total phenols and flavonoids by hot water extraction. They used a series of in vitro assays to test the antioxidant capacities of the extracts against various biologically relevant free radicals and oxidizing agents in psoriasis, revealing that, of the extracts obtained from maceration, yellow birch was the most effective at removing H_2_O_2_, HO^•^, and O_2_^•−^, in addition to BS from the extracts obtained with hot water extraction [99]. Agarwal et al. compared three different European tree barks with regard to their antioxidant properties. Via HPLC-PDA-ESI-MS/MS, they confirmed the presence of 123 different polyphenols potentially responsible for the antioxidant properties of the bark extracts. Interestingly, while some of the compounds were found in all of the extracts, most of them were specific to different tree species [94]. These findings clearly reveal the importance of extraction method and solvent choice when investigating the antioxidative activities of natural products.

In recent years, several studies have attempted to find natural sources of potent antioxidants that may have the potential to treat different types of chronic and degenerative diseases, including dermatological disorders. In order to investigate the suitability of plant byproducts as food antioxidants and preventative agents against skin aging as well as cancer, Touriño et al. generated fractions of different procyanidin compositions and structures from a total polyphenolic extract of maritime pine [100]. The main contents of the pine bark fractions were catechin monomers and other flavonoid monomers, mainly taxifolin, procyanidin dimers and oligomers, generated by RP-HPLC and Toyopearl HW-40. They used ABTS and DPPH assays to measure the free radical scavenging activity of the procyanidin fractions. Through an ABTS assay, all but one of the fractions showed more scavenger efficiency than the standard antioxidant Trolox. The results of the DPPH assay were similar to those obtained with ABTS, displaying the general trend that the higher the degree of polymerization, the higher the number of hydroxyls and therefore the free radical scavenging power per molecule. [100] Gascón et al. investigated the activities of different pine bark extracts on human colorectal adenocarcinoma (Caco-2) cells. They found different antioxidative components of the pine bark extracts, including taxifolin, catechin, and procyanidin B1 as well as B2, and showed for all of the investigated extracts a significant decrease in ROS generation in the Caco-2 cells after 72 h of incubation with 20 or 1000 µg/mL of pine bark extract, with maritime pine as the bark extract with the highest antioxidant capacity. Furthermore, maritime pine bark extract displayed the highest amount of procyanidin B2 and showed the highest biological activity, indicating that this component plays an important role. This finding has been confirmed by various studies [101,102,103]. The conclusion of the study by Gascon et al. was that the cytotoxic effect of the extracts could be related to disturbances in redox balance due to antioxidative properties, and that these could act synergistically to produce cell damage, cell cycle disruption, and intrinsic apoptosis induction by involving changes in matrix metalloproteinase (MMP), cytochrome c-release, and caspase 3-activation in Caco-2 cells. [78] Maritime pine is the species within the pine family that is the best-studied and most widely used in medicine. Two commercial products have been extracted from these pine trees growing on the west coast of France (Pycnogenol^®^ (Geneva, Switzerland)—PYC—and Flavangenol^®^ (Saga, Japan)—FLA) and widely studied, in vitro and in vivo [104,105,106,107]. While Flavangenol^®^ is obtained by hot water extraction, Pycnogenol^®^ is extracted with water and alcohol. Flavangenol^®^ consists of a concentrate of catechin, taxifolin, and proanthocyanidins, with procyanidin B1 as one major component. Among others, the extract showed antiphotoaging and anticarcinogenetic activities in melanin-possessing hairless mice, which may be due to a scavenging effect on ROS-inhibiting Ki-67, 8-OHdG, and VEGF expression [108]. Brizi et al. tested the neuroprotective activities of natural sweet chestnut extract on human neuroblastoma cells (SH-SY5Y) and demonstrated its potential ability to act as an antioxidant and neuroprotective agent. The tested extract significantly protected against oxidative stress through a concentration- and time-dependent decrease in H_2_O_2_-induced intracellular ROS formation, with maximal effects at 50 µg/mL (the highest tested concentration). The extract had a significant effect on cell viability, but showed no cytotoxicity at any concentration. Sweet chestnut extract reduced DNA damage, measured as nuclear condensation and DNA fragmentation, in neuroblastoma cells induced by glutamate, leading to the suggestion that a reduction in ROS might protect cells from DNA-damage-mediated apoptosis [109]. A further study investigated the effect of willow bark extract on human vascular endothelial cells [110]. HUVEC cells showed significantly less oxidative-stress-induced toxicity when previously treated with 100 µg/mL willow bark extract for 16 h. The effects of the extract on antioxidant enzymes were determined by the treatment of the cells with different doses (25–400 µg/mL), where in a dose-dependent manner the induction of antioxidant genes (HO-1, GCLM, GCLC, and p62) and their corresponding intracellular protein levels, as well as significantly increased intracellular GSH and Nrf2, could be demonstrated [110]. The role of transcription factor Nrf2 activation in the protective effects of willow bark extract was highlighted by this study. Oxidative stress triggers the activation of this redox-sensitive regulatory transcription factor [111]. Nrf2 regulates GSH levels, supports NADPH production (obligatory cofactor for many antioxidant systems), induces the expression of the ferritin complex genes (ROS detoxification) [112], and induces the expression of an array of further antioxidant response element-dependent genes. They also investigated if salicin, a major active component of willow bark extract (the extract used contained more than 15%), and its metabolites were the main contributors to Nrf2-mediated antioxidative activity by fractionating the extract. They could clearly show that fractions containing less to no salicin showed more intense activity compared to salicin-rich fractions, revealing that other active ingredients may contribute to the effects of willow bark extract on Nrf2 activation. This reveals the importance of investigating crude extracts and the synergistic effects of their components, not only focusing on isolated ingredients [110].

## 5. Oxidative Stress in Dermatology

The relationship between skin diseases and oxidative stress is well-researched, as excessive ROS are closely involved in the onset and development of numerous skin disorders [113], including skin cancer, cutaneous inflammation, autoimmunological processes, vasculitis, erythema, and edema [30,113], as well as in skin aging, including wrinkling and photoaging [114]. A study of the levels of blood superoxide dismutase, blood glutathione peroxidase, and serum MDA in 50 patients with melasma (a common acquired disorder of hyperpigmentation) showed a significant association between melasma and oxidative stress compared to a control group, leading to the assumption that oxidative stress is an important factor in the etiopathogenesis of melasma [115]. Oxidative stress has also been shown to play an important role in the pathogenesis of atopic dermatitis, a chronic relapsing inflammatory skin disease, by damaging dermal cellular structures, increasing skin inflammation and thereby weakening the skin barrier function in addition to enabling infections by microbial pathogens [34,116]. In acne vulgaris, a chronic inflammatory disease, oxidative stress may also play an important role. Elevated levels of MDA (lipid peroxidation product) and XO (xanthine oxidase), as well as reduced CAT (catalase) activity, all parameters of oxidative stress, have been found in the venous blood of patients suffering from this disease. During inflammation, xanthine dehydrogenase (which does not generate free radicals) present in normal tissues is converted to XO, a biological source of ROS. CAT plays a role in the defense against oxygen species by destroying H_2_O_2_. Increased XO activity and lipid peroxidation, as well as decreased CAT activity, may play a role in relation to increased oxidative stress, leading to the etiopathogenesis of acne vulgaris [117]. The list of dermatological diseases associated with significant increases in oxidative stress also includes, but is not limited to, contact dermatitis [118], seborrheic dermatitis [119], alopecia areata [120], and rosacea [121]. DNA lesions, protein conversion, and lipid peroxidation, as well as the alteration of the activities of specific signaling pathways, such as Nrf2, NF-κB, MAPK, PI3K/Akt, and overacting cytokines/immune mediators, are outcomes of oxidative stress, which can promote the occurrence and aggregation of dermatoses [122]. Nrf2 plays an important role in the protection of skin cells against environmental oxidative stress by controlling the antioxidant response system. Phytochemicals have the ability to mitigate UVR-induced skin damage, such as photoaging, inflammation, and melanogenesis, as well as prevent photocarcinogenesis, amongst others, via the activation of Nrf2 signaling-regulated redox balance [123].

Plant extracts have been used for skincare purposes for centuries [124], and are still very popular in skincare, cleaning, and protection. Knowledge on the antioxidant activities of bark extracts could be essential for the development of new drugs and dietary supplements. Malonic acid isolated from Japanese pine (*Pinus densiflora*) increased the levels of the antioxidant enzymes SOD-1 and HO-1 (heme oxygenase) via the activation of Nrf2 in a human keratinocyte cell line (HaCaT), thereby reducing UVB-induced ROS levels [125]. After the irradiation of human dermal fibroblasts with UVA, two isolated compounds of the water extract from the *Galinsoga sp.* herb (caffeic acid derivates 2,3,5(2,4,5)-tricaffeoylaltraric acid and 2,4(3,5)-dicaffeoylglucaric acid) were shown to increase HO-1 expression and activate the Nrf2 transcription factor, thereby demonstrating their possible protection against UVA-induced oxidative stress [126]. The supplementation of 75 mg of Pycnogenol^®^ daily over a 30-day clinical trial led to a decrease in the average melasma area and the average pigmentary intensity in 30 women with melasma, showing a general effective rate of 80%, without displaying any side effects. Ni et al. concluded that Pycnogenol^®^ was a therapeutically effective and safe treatment in patients suffering from melasma [127]. Another clinical trial that tested the oral supplementation of Pycnogenol^®^ with 1.10 mg/kg body weight for 4 weeks and 1.66 mg/kg body weight for 4 more weeks showed a dose-dependent significant increase in the mean minimal erythema dose (a reliable parameter of UV-exposure-induced acute inflammatory reactions of the skin) in 21 fair-skinned volunteers. UVR-induced ROS enhancement and oxidative damage are proposed to stimulate NF-κB activation. This pro-inflammatory and redox-regulating transcription factor is thought to be activated in UVR-induced erythema in human skin and was used as a marker of the proinflammatory response of keratinocytes after UV exposure. Pycnogenol^®^ was further studied in HaCaT cells to determine its effect on NF-κB. UVR-mediated NF-κB-dependent gene expression was significantly decreased in a dose-dependent manner through the addition of Pycnogenol^®^ to the medium. This was thought to be due to the free radical scavenging and transition metal chelation properties of pine bark extract, as well as the fact that the inhibitory effect on NF-κB-dependent gene expression might lead to the observed increase in MED [106]. Additionally, black spruce bark extract was shown to down-regulate the NF-κB pathway in TNF-α-activated psoriatic keratinocytes [128]. Birch bark extract improved skin barrier health and exhibited a strong restructuring effect and corneocyte cohesion of the skin at a very low concentration (formulation containing 0.1% extract) in a 28-day treatment of 35 volunteers with dry/very dry skin [129].

These are promising results, but so far only a limited number of tree bark species, especially European trees, have been investigated for their antioxidative properties. Further knowledge will be fundamental for the development of new dietary supplements and pharmaceutical products that provide protection against harmful exogenous or endogenous factors and thus prevent or reduce the effects of skin disorders caused by oxidative stress.

## 6. Antimicrobial Effects of Tree Bark Extracts

Due to the intrinsic protection of plants against a wide diversity of invaders, including bacteria and fungi, plants and their constituents have attracted attention in terms of their usage as antimicrobial agents over decades. Nowadays, as bacterial resistance to common antibiotics increases, the search for natural effective antibiotics intensifies [130]. Considerable contributions to the knowledge of the antimicrobial actions of phytochemicals come from the research on herbal plants documented by the number of publications in the field of phytochemicals for medical applications, which dates back to the time of Hippocrates, around the 5th century B.C. [131]. Within the European Union, a committee on Herbal Medicinal Products under the authority of the European Medicines Agency (EMA) has been established and is responsible for compiling and assessing scientific data on herbal substances, preparations, and combinations. The EMA provides a list of recommended herbal medicinal products with antibacterial activity in the context of skin and wound infections. A number of well-known medical plants from European folk medicine are listed, among them the bark extract of Pedunculate oak for the symptomatic treatment of minor inflammation of the oral mucosa or skin [34]. Common oak being the only bark extract listed highlights the lack of knowledge on the therapeutic potential of wood bark extracts, even though the bark of common temperate trees contains a number of secondary metabolites potentially beneficial for various indications in the medical field.

The main constituents of plant-derived compounds responsible for their antimicrobial effect include phenolic compounds, terpenoids, alkaloids, carotenoid, and sulfur-containing phytochemicals [132]. Polyphenols are the group of compounds best characterized for their antimicrobial action and are divided into two major classes: flavonoids and non-flavonoids/phenolic acids. Even though research relating to the relationship between the structures and antimicrobial properties of polyphenols is ongoing, a clear structure–activity relationship has not yet been established [133]. The antibacterial modes of action of polyphenols include, among others, the interaction with a cytoplasmatic membrane [134], influence on the virulence factor [135], inhibition of bacterial topoisomerase activity [136], direct action on the bacterial cell wall [137], and inhibition of bacterial gene expression [138]. These mechanisms are among the main targets for antibiotics, which in general include bacterial lipid membrane integrity, proteosynthesis, bacterial mRNA synthesis, bacterial cell wall synthesis, folic acid synthesis, transmembrane potential, ATP production, efflux pump, and the synthesis as well as integrity of bacterial DNA [34].

Dissenting opinions exist regarding the use of isolated compounds versus crude extracts as antimicrobial agents. In this review we focus on the use of crude extracts, with the notion that through the diversity of the compounds within each extract a number of targets are affected, leading to the destruction of the cellular integrity of bacteria. The main assays used to determine the antibacterial activity of plant extracts are the agar diffusion assay and the broth growth test [139]. The agar diffusion assay determines the extent of growth inhibition by measuring the diameter of inhibition zones on microbial-inoculated agar plates. The broth microdilution assay determines the number of metabolically active cells in the suspension [140].

Oak is one of the best characterized European tree bark extracts due to its biological activity in terms of antioxidant capacity as well as antimicrobial, antiproliferative, immunomodulatory, and hypoglycemic effect [141]. Recent studies investigating the antimicrobial effect of oak bark extracts using various extraction methods confirmed their antimicrobial potential. Comparing the ultrasonic-assisted extraction and microwave-assisted extraction (MAE) of northern red oak revealed a higher phenolic content in the MAE extracts, which correlated with a higher antibacterial and antifungal potential, especially against *Staphylococcus aureus* (*S. aureus*), *Candida parapsilopsis* (*C. parapsilopsis*), and *Candida krusei* (*C. krusei*) [142]. Another study that investigated the antimicrobial activity of oak bark (*Quercus robur*) used a step gradient elution with different polarities of solvents by ExtraChromR to obtain six fractions characterized by different polarities and lipophilicities. Interestingly, *S. aureus* could only be inhibited by highly polar extracts, whereas less polar extracts showed antimicrobial activity against the Gram-negative bacteria *Enterobacter aerogenes* (*E. aerogenes*) and the yeast *Candida albicans* (*C. albicans*). The active substances against *E. aerogenes* and *C. albicans* could be found in lipophilic extracts, whereas substances in lipophobic extracts were active against *S. aureus* [143]. Another study focusing on the influence of the polarities of extracts in relation to antimicrobial effects showed that the non-polar extracts of Norway spruce exhibit a strong response against most of the Gram-positive bacteria tested, while Gram-negative bacteria presented high resistance to almost all of the extracts, with *K. pneumoniae* being the most susceptible [144]. Ferrentino et al. investigated the effect of different extraction methods, Soxhlet and supercritical carbon dioxide, and determined the phenolic compounds responsible for the antimicrobial action of spruce. Regardless of the technology used, both extracts showed similar antimicrobial activity against *Streptococcus thermophilus* (*S. thermophilus*) and *Enterococcus faecalis* (*E. faecalis*). Phenolic compounds identified to be responsible for the antimicrobial effect include catechin, dihydroquercetin, astringin, and isorhapontin [145]. The relationship between the polyphenolic contents and antibacterial activities has been further investigated using wild cherry, European larch, and sweet chestnut tree bark extracts on *S. aureus* and *Escherichia coli* (*E. coli*). Bacterial growth curves showed that the extracts had no antibacterial activity against *E. coli*; in contrast, the extract supported their growth, whereas a significant inhibitory effect of the extracts was observed on *S. aureus*. Sweet chestnut was the most effective extract that can be linked to the highest antioxidant potential and highest total phenol content compared to wild cherry and European larch [94]. Another study investigating the correlation between the scavenging ability of superoxide radicals and the antimicrobial activity of sweet chestnut revealed a high antimicrobial activity against *Micrococcus pyrogenes* var. *albus*, *S. aureus*, and *Salmonella typhimurium* (*S. typhimurium*), which correlated significantly with phenol and flavonoid content [146]. Common beech bark extracts isolated using pure hot water extraction were active against the growth of *S. aureus* and methicillin-resistant *S. aureus* (MRSA), which could be linked to polyphenol content, including vanilic acid, catechin, taxifolin, and syringin, whereas the effect of polyphenolic extract on Gram-negative bacteria was absent at a concentration of 30 mg/mL beech bark extract [147]. The trend that the growth inhibition of Gram-positive bacteria is stronger than in Gram-negative bacteria has been observed in previous studies. Emrich et al. investigated the antimicrobial effects of birch, beech, and larch bark extracts in MRSA (methicillin-resistant Staphylococcus aureus), *E. coli*, *Cutibacterium acnes* (*C. acnes*), and *Staphylococcus epidermidis* (*S. epidermidis*) strains, and showed that all bark extracts could inhibit the growth of Gram-positive bacteria; however, Gram-negative *E. coli* could not be inhibited. The effect of larch bark extract was minor compared to that of birch and beech in relation to growth inhibition [81]. Similar results were shown by Laireiter et al., who investigated the antimicrobial effects of larch and pine wood on four bacterial strains. Pine heartwood inhibited the growth of Gram-positive *S. aureus*, *Bacillus subtilis* (*B. subtilis*), and *E. faecium*; however, Gram-negative *Pseudomonas aeruginosa* (*P. aeruginosa*) could not be inhibited. In addition, substantial differences could be observed when investigating different parts of trees, with the sap-, heart-, and knotwood of larch trees being inactive against the tested bacteria, with only the bark of larch being effective against *S. aureus* [148]. Isolated compounds from birch bark extracts showed that oleanolic acid demonstrated prominent antibacterial activity against *S. aureus* and *B. subtilis*; however, *E. coli* could not be inhibited [149]. A comprehensive study looking at the antimicrobial activity of Pycnogenol^®^ against 23 different pathogenic prokaryotic (Gram-positive and Gram-negative) and eukaryotic (yeast and fungi) microorganisms showed that all of the tested species could be inhibited using minimum concentrations ranging from 20 to 250 µg/mL [150].

In summary, these studies showed that European tree bark extracts have great potential as an abundant source of bioactive compounds with antimicrobial properties. Various solvents and extraction methods can be used to tailor extracts to enrich certain compounds in order to target specific pathogens. A clear relationship between the polyphenolic content of the extracts of woody vascular plants and their antimicrobial activity was observed [142]. However, there is still a limited amount of information on European tree bark extracts compared to plant-derived extracts investigated for their use as potent drug candidates against various pathogens, including antimicrobial-resistant pathogens [151]. For some abundant trees that are highly utilized in the wood industry, such as maples and beech, very limited information is available regarding their biological activities [29,81].

In light of antimicrobial activity in the context of skin and wound infections, the use of plant-derived therapeutic agents has a long tradition, as listed by the number of plant extracts registered as natural products for therapeutic use for skin [34]. The composition of skin microbiota varies depending on the physiology of the skin site, with moist, dry, and sebaceous microenvironments. The most abundant bacteria in healthy individuals include *Propionibacterium*, *Staphylococcus*, and *Corynebacterium* species, which are in constant exchange with eukaryotic and viral species present at various physiological sites on the skin. These help to prevent colonization by pathogenic bacteria [152]. Many common skin diseases, such as acne, eczema, and chronic wounds are characterized by a change in the skin microbiota. The chronic inflammatory skin condition common acne is an infectious disease that affects about 85% of teenagers and is characterized by scaly red skin, pinheads, blackheads and whiteheads, and large papules [153]. Acne vulgaris is a multifactorial disorder that involves sebaceous hyperplasia, follicular hyper keratinization, and colonization by *C. acnes*, acting as an opportunistic pathogen. The imbalance in skin microbiota is suggested to promote the selection of pathogenic strains of *C. acnes* [154]. However, the extensive use of antibiotics in acne vulgaris has led to an induction of antibiotic resistance in up to 40% of *C. acnes* strains, leading to an increased likelihood of treatment failure [155]. In the light of the search for alternative antimicrobial agents that act against *C. acnes*, a study by Emrich et al. showed a strong inhibitory action of birch and beech bark extracts [81]. Similar issues regarding the use of broad-band antibiotics concern therapy for atopic dermatitis, for which an abundance of *S. aureus* has been correlated to the severity of atopic dermatitis. The aim would be to develop novel therapies specific for *S. aureus* to avoid the administration of broad-band antibiotics. A number of studies, as described above, have investigated the effect of wood bark extracts on *S. aureus* that show a promising inhibition of growth, which should be taken into consideration for the search of alternative treatment options [81,94,142,143,147,148]. Another study, which investigated the dermo-cosmetic effect of common temperate trees, screened 30 wood bark extracts via bioautography against *S. aureus*. Results showed that methanol extracts of pedunculate oak and European larch bark were the most potent, followed by water extracts of oak, larch, and spruce bark [29].

Another strategy with which to fight against increasing antimicrobial resistance is the use of combination therapies of herbal drugs and phytochemicals, with antibiotics co-administrated, to act synergistically in reversing resistance mechanisms [133,156]. However, in the field of combination therapy the available literature is based on isolated compounds or extracts as herbal drugs, with limited information regarding the use of woody vascular plants.

Therefore, great potential lies in the exploration of woody vascular plants as drug discovery sources in the battle against bacteria, especially as these natural products act comprehensively by exerting not only antibacterial effects but also anti-inflammatory and potential supportive effects in tissue regeneration and wound healing.

## 7. Wound Healing and Immune-Regulating Capacity of Tree Bark Extracts

Natural products and plant extracts have been widely used as topical applications for wound healing and skin regeneration in traditional medicine, and ethnobotanical knowledge in this field has been recorded over centuries in folklore medicine [157]. The process of wound healing is characterized by four time-dependent steps, including hemostasis, an inflammatory phase, a proliferative phase, and a remodeling phase [158]. Immediately after injury, the formation of blood clots prevent exsanguination from vascular damage, which follows a stringent mechanism to avoid excessive thrombosis. During the inflammatory phase, the innate immune system plays a central role in the defense against pathogenic wound invasion. Damage-associated molecular patterns and pathogen-associated molecular patterns activate resident immune cells, releasing chemo attractants and cytokines, leading to vasodilation and the further recruitment of immune cells, such as neutrophils and macrophages [158]. In addition, one of the first responses of the innate immune system is the production of ROS to initiate tissue repair [159]. The third phase encompasses the proliferation and migration of fibroblasts as well as collagen synthesis and epithelialization. The final stage in the wound healing process aims to achieve maximum tensile strength through reorganization, degradation and synthesis of the extracellular matrix [158].

Plant-based extracts can support tissue regeneration through a variety of mechanisms that target different aspects of the wound healing process [160]. There are a number of medicinal plant materials and herbal remedies that are documented to play a significant role in curing acute and chronic wounds [158]. Plant-derived phytochemical compounds with wound healing properties include, among others, alkaloids, cardenolides, coumarin, glucoside, polyphenols, sterols, tannins, and terpene [161]. Among the group of polyphenols, flavonoids are a group of compounds with promising wound healing properties that act through a variety of mechanisms [159]. A review by Carvalho summarized the latest literature on the beneficial effects of flavonoids in the regulation of inflammation as well as MMPs, and further states their angiogenic activities that promote sustained vascularization and oxygen supply [159]. It has been shown that flavonoids have important anti-inflammatory properties by inhibiting regulatory enzymes or transcription factors involved in inflammation, leading to a decrease in a number of inflammatory mediators [162]. In addition, the overexpression of NFĸB, which contributes to failure in the healing process, is regulated by flavonoids [163]. MMPs play crucial roles in all steps of the wound healing process, whereby a tight balance of the level of MMPs is needed to allow fast tissue regeneration and prevent poor wound healing as well as chronic wounds [164]. Another mode of action by which polyphenols can promote wound healing is through their antimicrobial and free radical scavenging abilities, which prevent cell damage, as has been discussed in detail in the previous chapters.

Many of the compounds described above can be found in wood bark extracts of common temperate trees, and new emerging studies are investigating these extracts in relation to wound healing, highlighting their potential. A prime example is the clinically proven wound healing efficacy of birch bark extract with its active ingredient betulin, a pentacyclic triterpene. Episalvan^®^ (Niefern-Öschelbronn, Germany), a birch bark extract with betulae cortex as the main active substance, received approval by the EMA for the treatment of superficial skin wounds and IIa-degree skin burn wounds in adults in 2016; however, it was withdrawn by the European union as a medicine in 2022 [165,166]. A new medicine, Fisuvez^®^ (Niefern-Öschelbronn, Germany), a dry extract from birch bark using n-heptane as the extraction solvent, has recently been approved by the EMA for the treatment of epidermolysis bullosa [23]. It has been shown that triterpenes act through the regulation of arachidonate metabolism to transiently initiate a controlled inflammatory response, promote keratinocyte migration by increasing the formation of actin filopodia, lamellipodia, and stress fibers, and contribute to the clearance of a wound through their antimicrobial mode of action [20,167]. Ebeling et al. elucidated the molecular mechanisms of betulin as well as birch bark extract (triterpene extract) and showed a transient upregulation of pro-inflammatory cytokines, chemokines, and cyclooxygenase-2, in addition to the promotion of keratinocyte migration through the activation of Roh GTPases, which was confirmed in scratch assay experiments with primary human keratinocytes and in a porcine ex vivo wound healing model (WHM) [20]. In addition, triterpenes are known to improve scar formation for superficial lesions and have recently been approved by the EMA as agent for the treatment of *Epidermolysis bullosa dystrophica* and *junctionalis*, suggesting a faster re-epithelialization of wounds [23,168,169]. Betulin is known to regulate inflammation and showed promising results in the reduction in inflammation in an ear oedema model [170]. A pilot study using birch bark extract for the treatment of actinic keratosis showed important therapeutic activity by effectively reducing lesions via an anti-inflammatory mechanism [171].

Oak bark extract is another well-known and commonly used traditional medical plant in the areas of skin infections and wound healing. A study that evaluated the wound healing effect of ethanolic bark extract of evergreen oak on full-thickness excisions in Wistar rats revealed an accelerated healing process due to accelerated wound contraction and a reduced epithelialization period compared to the used positive control, Cicatryl-bioR [172]. The positive effects can be, among others, related to the high contents of flavonoids, including alpha-tocopherol, which promote the formation of a new epithelium by increasing the migration of keratinocytes through the activation of the Ras/Raf/MEK/ERK pathway and the increased expression of MMP-9 [159]. In addition, the presence of phenolic compounds, such as gallic acid and paracoumeric acid, acts as an antioxidant, which is crucial to maintain a homeostatic balance between antioxidant enzymes and ROS, which is crucial in the wound healing process [159].

Research in the field of biomedical textiles for application in skin protection and wound healing aims to design products that provide accelerated wound healing, antimicrobial activity, and protection against oxidative stress. Bark extract of Turkish pine was compared against Pycnogenol^®^ to evaluate its antimicrobial and wound healing properties as a formulation applied to cotton fabrics [173]. Turkish pine showed advanced antimicrobial activity against *Aspergillus brasiliensis* as well as increased keratinocyte cell proliferation and accelerated cell-free gap closure compared to Pycnogenol^®^. Bark extracts from various pine species have been reported to accelerate the wound healing process by enhancing capillary permeability [174], exerting an anti-inflammatory activity [175] and protecting against reactive oxygen and nitrogen species [176] linked to high contents of polyphenols, including proanthocyanidins, catechin, and taxifolin [177]. In light of the utilization and sustainability of byproducts in the woodworking industries, common beech bark extracts were investigated in terms of their wound healing capacity and immune modulation. In vitro analyses using the human HaCaT keratinocyte cell line showed accelerated wound closure at low concentrations of beech bark extracts, correlating with a slight but non-significant increase in the inflammatory cytokines IL-8 and IL-1 beta [81]. Investigations concerning dermo-cosmetic properties, including the capacity to inhibit tyrosinase, elastase, and collagenase activity, have been performed on isolates of Norway spruce and bird cherry bark extracts. Fractions of methanol-soluble metabolites of common spruce bark corresponding to stilbene, flavonoid, and phenolic acid derivatives showed important anticollagenase and antimicrobial activities. The compounds E-piceid and taxifolin exhibited significant antityrosinase activity [178]. Bird cherry bark extract obtained by hot water extraction exhibited moderate elastase inhibition and mild tyrosinade inhibition compared to standards for each assay; however, very high polyphenol concentrations make bird cherry bark extract a potent antioxidative agent for cosmetic applications [32].

Even though there is a growing number of studies focusing on the potential benefit of phytochemicals in the prevention and treatment of wounds, the literature on bark extracts of European woody vascular plants, apart from on the more characterized bark extracts of birch, oak, and pine, is very limited. However, summarizing the available literature suggests a combinatorial effect of antimicrobiotic, antioxidant, as well as immunomodulating modes of action that might be among the contributing factors that lead to remarkable improvement in the wound healing process of certain wood extracts for various skin indications (Table 3).

## 8. Conclusions

In summary, bark extracts derived from various deciduous and coniferous tree species in Europe, discussed in this review, showed promising potentials as antioxidant, antimicrobial, wound healing, and immune-regulating agents. As depicted in Table 3, a number of bark extracts show synergistic effects by inhibiting various—mainly Gram-positive—bacteria and exhibiting strong radical scavenging ability, determined in both chemical and biological assays. In addition, mainly birch and oak bark extracts can significantly accelerate dermatological wound closure and are therefore used for various skin indications. In view of their potential health benefits, there is an ongoing search for natural compounds that can promote skin health and protect skin cells from pathological processes mediated by oxidative stress, dysregulated immune function, and microbial colonization. This review clearly highlights the potential benefits from tree bark extracts for various indications; however, it also illustrates the need for the further investigation of different tree species to identify their full potential. 

## Figures and Tables

**Table 1 antibiotics-12-00130-t001:** Botanical classification of common European tree species that are discussed in this review.

Order	Family	Genus	Species	Common Name
Fagales	Betulaceae	*Alnus*	*Alnus glutinosa*	Black alder
Fagales	Fagaceae	*Fagus*	*Fagus sylvatica*	Common beech
Fagales	Fagaceae	*Quercus*	*Quercus robur*, *Quercus rubra*, and *Quercus ilex*	Pedunculate oak, northern red oak, and evergreen oak
Rosales	Rosaceae	*Prunus*	*Prunus avium*, *Prunus padus*	Wild cherry, bird cherry
Sapindales	Sapindaceae	*Acer*	*Acer pseudoplatanus*	Sycamore maple
Lamiales	Oleaceae	*Fraxinus*	*Fraxinus excelsior*	European ash
Coniferales	Pinaceae	*Larix*	*Larix decidua*	European larch
Coniferales	Pinaceae	*Picea*	*Picea abies*	Norway spruce
Fagales	Betulaceae	*Betula*	*Betula pendula*	European white birch
Coniferales	Pinaceae	*Pseudotsuga*	*Pseudotsuga menziesii*	Douglas fir
Coniferales	Pinaceae	*Abies*	*Abies alba*	European silver fir
Malpighiales	Salicaceae	*Salix*	*Salix alba.*	Willow
Pinales	Pinaceae	*Pinus*	*Pinus pinaster*, *Pinus brutia*	Maritime pine, Turkish pine
Fagales	Fagaceae	*Castanea*	*Castanea sativa*	Sweet chestnut

**Table 3 antibiotics-12-00130-t003:** Synergistic effects of wood bark extracts. Assays for which a significant effect was detected (scavenging ability and wound healing) are listed.

Species	Main Active Compounds	Antioxidative Activity *	Antimicrobial Activity	Wound Healing Assay	Skin Indications/Cosmetic Products	Ref.
Beech bark	Flavonoids (catechin, epicatechin, quercetin, taxifolin, and procyanidins), phenolic acid (protocatechuic acid), syringic acid, gallic acid, and hydroxycinnamic acids (chlorogenic acid, coumaric acid)	DPPH	*S. aureus*, MRSA, *S. epidermidis*, *C. acnes*, *E. coli*, *P. aeruginosa*, *S. typhimurium*, *C. albicans*, *C. parapsilosis*, and *C. zeylanoides*	In vitro scratch assay keratinocytes and human melanoma cell line	Unknown	[81,142,179,180]
Birch bark	Pentacyclic triterpenes, hydrocarbons	DPPH	*S. aureus*, MRSA *S. epidermidis*, and *C. acnes*, *B. subtilis*	In vitro scratch assay keratinocytes, porcine ex vivo wound healing model, and ear edema model	Superficial skin wounds, IIa-degree burn wounds, epidermolysis bullosa (Filsuvez^®^), actinic keratosis, and cosmetic products	[20,21,81,149,166,168,169,170,171,181,182]
Pine bark	Flavonoids (proanthocyanidins, catechin, and taxifolin)	DPPH, ABTS, ORAC, SOD, and FRAP; decrease in NF-kB-dependent gene expression	*A. brasiliensis*, *Salmonella sp.*, *S. aureus*, *E. faecalis*, *E. coli*, *K. pneumoniae*, *P. mirabilis*, *P. aeruginosa*, *C. perfringens*, *Campylobacter sp.*, *S. glucans*, *B. cereus*, *C. albicans*, *A. oryzae*, *P. funculosum*, *F. monilifoorme*, *S. mutans*, *L. acidophilus*, *A. actinomycetemcomitans*, *Candida species*, and *moulds*	In vitro scratch assay keratinocytes	Psoriasis; melasma (decrease in melasma area and pigmentary intensity); erythma; pine-extract-reduced susceptibility of skin and isolated HaCaT cells to UV-R exposure	[86,106,127,148,150,173,174,175,176,177,183]
Oak bark	Flavonoids (alpha-tocopherol), phenolic acids (gallic acid), paracoumeric acid, and tannins	DPPH, ABTS, ORAC, SOD, and FRAP	*S. aureus*, *C. parapsilopsis*, *C. krusei*, *E. aerogenes*, and *C. albicans*	In vivo full-thickness excisions in Wistar rats	Unknown	[12,13,14,15,29,80,86,141,142,143,172]
Larch bark	Flavonoids, lignans, and stilbenoids (astringin)	DPPH, ATBS, and FRAP	*S. aureus*, MRSA, *C. acnes*, and *S. epidermidis*	Not determined	Cosmetic products	[29,81,94,148]
Spruce bark	Stilbene (E-piceid), flavonoids (taxifolin), phenolic acid, tannins, and lignins	DPPH	*C. albicans*, *C. tropicalis*, *A. baumannii*, *L. monocytogenes*, *E. faecalis*, *B. cereus*, *S. thermophilus*, and *S. aureus*	Scratch assay human melanoma cell line (A375)	Anti-wrinkling (inhibit tyrosinase, elastase, and collagenase); cosmetic products	[29,144,145,178,179]
Cherry bark	Flavonoids (mainly flavanones and flavonols) and flavonoid glycosides	DPPH, ATBS, and FRAP	*S. aureus*	Not determined	Anti-wrinkling (inhibit tyrosinase, elastase, and collagenase); cosmetic products and food supplements	[32,94,184]
Sweet chestnut bark	Tannins (gallotannins and ellagitannins), flavonoids	DPPH, ATBS, and FRAP; human neuroblastoma cells protected against oxidative stress through reduced DNA damage by glutamate	*S. aureus*, *M. pyrogenes*, and *S. typhimurium*	Not determined	unknown	[94,109,146]
European ash	Esculetin, esculin, fraxin, and fraxetin	DPPH, ABTS, and FRAP		Not determined	Unknown	[94]
Willow bark	Flavonoids, salicylates	Induction of antioxidative genes and increased GSH levels in HUVEC cells	*S. aureus*, *P. aeruginosa*, *C. albicans*, and *E. coli*	Not determined	Cosmetic products (anti-aging, anti-wrinking)	[110,185,186]

* Antioxidative assays for which a significant an effect was detected.

## Data Availability

Not applicable.

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
