# Peer review of "Antioxidative and Antimicrobial Evaluation of Bark Extracts from Common European Trees in Light of Dermal Applications"

_antibiotics, 2023, doi:10.3390/antibiotics12010130_

Round 1

Reviewer 1 Report

Congratulations for your work. The interesting topic of the paper falls within the scope of MDPI Antibiotics.

The proposed hypotheses are important with high scientific soundness.

The experimental and analysis methodology are suitable with high feasibility.

You have highlighted the aims, significance and the novelty of your work.

Please change the text:

An antioxidant is “any substance that delays, prevents or removes oxidative damage to a target molecule” [73], therefore playing an important role in the protection against the consequences of oxidative stress. Rows 285-287

It is similar to: https://www.intechopen.com/chapters/66259

Please change the text:

The ratio of reduced to oxidized glutathione within cells is often used as a marker of cellular toxicity. Rows 342-343

It is similar to: https://www.ncbi.nlm.nih.gov/pmc/articles/PMC3506742

Reviewer 2 Report

Overall: In this review, the authors present current understanding and summary of bark extracts from European trees. Biological effects of bark extracts and the potential medical use are well discussed. The authors’ contribution is well covered and organized.

There are no major concerns about this article. I have few minor comments. 

-Line 54-55, relevant reference(s) is needed here for the introduction of oak.

-Line 134-135, relevant reference(s) is needed here for the proportion of fir (Abies alba).

For Section “3. Extraction technologies in relation to and relevance for biomedical applications”, it’s a little bit lengthy for the extraction technologies here,  I suggest briefing the introduction of extraction technologies and focusing more on the biomedical relevance to highlight the applications.

Line 428-430, what type of DNA damage did sweet chestnut reduce? DNA double-strand breaks or other kinds of damage?

Line 430-437, relevant reference(s) is needed here.

Line 476-482, the authors may consider moving the introduction of Nrf2 to the end of Section 4, where Nrf2 is described first in the manuscript.

Line 593-594, full names of S. aureus and E. coli need to be given for the first time they are shown in the manuscript, same as the other bacteria.

In Table 3, I suggest writing “unknown” or “not determined” for the blank position, eg. blank of “Skin indications/cosmetic products” for “Beech bark”. 
